# Current Evidence on the Efficacy of Gluten-Free Diets in Multiple Sclerosis, Psoriasis, Type 1 Diabetes and Autoimmune Thyroid Diseases

**DOI:** 10.3390/nu12082316

**Published:** 2020-08-01

**Authors:** Moschoula Passali, Knud Josefsen, Jette Lautrup Frederiksen, Julie Christine Antvorskov

**Affiliations:** 1Multiple Sclerosis Clinic, Department of Neurology, Rigshospitalet-Glostrup, Valdemar Hansens Vej 13, 2600 Glostrup, Denmark; passali@nexs.ku.dk; 2Department of Clinical Medicine, Faculty of Health Sciences, University of Copenhagen, Blegdamsvej 3B, 2200 Copenhagen N, Denmark; 3The Bartholin Institute, Rigshospitalet, Ole Maaløes Vej 5, 2200 Copenhagen, Denmark; knud@eln.dk (K.J.); julie.antvorskov@gmail.com (J.C.A.)

**Keywords:** gluten, gluten-free diet, gliadin, autoimmunity, neurology, multiple sclerosis, psoriasis, autoimmune thyroid disease, type 1 diabetes, celiac disease

## Abstract

In this review, we summarize the clinical data addressing a potential role for gluten in multiple sclerosis (MS), psoriasis, type 1 diabetes (T1D) and autoimmune thyroid diseases (ATDs). Furthermore, data on the prevalence of celiac disease (CD) and gluten-related antibodies in the above patient groups are presented. Adequately powered and properly controlled intervention trials investigating the effects of a gluten-free diet (GFD) in non-celiac patients with MS, psoriasis, T1D or ATDs are lacking. Only one clinical trial has studied the effects of a GFD among patients with MS. The trial found significant results, but it is subject to major methodological limitations. A few publications have found beneficial effects of a GFD in a subgroup of patients with psoriasis that were seropositive for anti-gliadin or deamidated gliadin antibodies, but no effects were seen among seronegative patients. Studies on the role of gluten in T1D are contradictive, however, it seems likely that a GFD may contribute to normalizing metabolic control without affecting levels of islet autoantibodies. Lastly, the effects of a GFD in non-celiac patients with ATDs have not been studied yet, but some publications report that thyroid-related antibodies respond to a GFD in patients with concomitant CD and ATDs. Overall, there is currently not enough evidence to recommend a GFD to non-celiac patients with MS, psoriasis, ATDs or T1D.

## 1. Introduction

Wheat is a major component of Western diets, however, abstaining from gluten is becoming a popular trend [1]. Adhering to a lifelong gluten-free diet (GFD) is the current treatment for celiac disease (CD)—an immune-mediated small intestinal enteropathy triggered by the ingestion of gluten [2]. It has been hypothesized that gluten may contribute to deteriorating the course of immune-mediated disorders [3,4,5]. According to a U.S. national survey, a GFD was the most common special diet to be used by patients with psoriasis [6]. Similarly, an American dietary survey found that 5.6% of the surveyed patients with multiple sclerosis (MS) reported adhering to a GFD [7], whereas in an Australian survey, a GFD was adopted by 16.4% of the included patients with MS [8]. Type 1 diabetes (T1D) and autoimmune thyroid diseases (ATDs) affect the endocrine system. The contribution of dietary factors to the pathogenesis of autoimmune endocrine disorders is currently an active research area. This review summarizes the currently available clinical data on a potential involvement of gluten in MS, psoriasis, T1D and ATDs.

## 2. Gluten

Gluten proteins have long been of interest to the food industry due to their high impact on the baking quality of wheat flours [9,10]. From a chemical perspective, gluten has been defined as the proteinaceous mass that remains when wheat dough is washed with water and consists primarily of the prolamin and glutelin fractions of the storage proteins of wheat [11,12]. The terms prolamin and glutelin originate from the classification of grain proteins into four fractions according to their solubility properties (Osborne fractions) [13]. Prolamins are insoluble in water but soluble in alcohol, whereas glutelins are insoluble in both water and alcohol [14]. The terms gliadin and glutenin account for the prolamin and glutelin fractions of wheat, whereas the terms secalin, hordein and avenin describe the prolamin fraction of rye, barley and oats, respectively [14]. Likewise, the glutelin fractions of rye and barley are commonly described as secalinin and hordenin, however, similar terminology does not apply for oat glutelins [12]. Codex Alimentarius has defined gluten as “a protein fraction from wheat, rye, barley, oats or their crossbred varieties and derivatives thereof, to which some persons are intolerant and that is insoluble in water and 0.5M NaCl” [15]. As a result, gluten is nowadays considered to be a common term for the prolamin and glutelin fractions of wheat, rye, barley and, in some cases, oats. 

Gluten proteins contain repetitive sequence sections that are rich in the amino acids proline and glutamine [14,16]. Such sections cannot be fully degraded by the human gastrointestinal enzymes [10,13], resulting in the presence of relatively long gluten peptides in the small intestine. In patients with CD, such gluten peptides trigger an inflammatory reaction, however, their presence in the small intestine of most healthy individuals is believed to be rather unproblematic. In vitro studies using caco-2 cell lines [17,18] as well as ex vivo studies on human biopsy explants from both CD patients and healthy controls (HCs) [18,19], suggest that exposure to gliadin disrupts the integrity of the intestinal epithelium. The effect of gliadin on intestinal permeability is believed to be mediated through the secretion of the protein zonulin [20]. Zonulin has been identified as prehaptoglobin-2 [21] and serum zonulin is often used as a marker of intestinal permeability. Levels of zonulin have been found to be elevated in autoimmune diseases [22,23,24,25], however, widely used ELISA kits cross-react with proteins, such as properdin and complement C3 [26,27], which shows why caution should be practiced when interpreting data on this topic.

## 3. Multiple Sclerosis

Multiple sclerosis (MS) is an autoimmune, yet incurable, disease of the central nervous system [28] and one of the leading causes of disability among young adults. A recent publication reports that 31% (10/32) of websites providing MS-specific dietary advice recommend patients with MS to abstain from “grains (gluten)” [29]. Despite a high interest in the use of dietary modifications to ameliorate the course of the disease [30], MS-specific, evidence-based dietary guidelines have not been developed yet. 

### 3.1. Gluten-Free Interventions in Multiple Sclerosis

The effects of a GFD among patients with MS have only been investigated by a single open label, non-randomized, controlled trial. Thirty-six patients, who followed a GFD for a median of 4.5 years (mean 5.3 ± 1.6), were compared with 36 patients who followed a regular diet [31]. At the end of the study, the group on the GFD had significantly lower disability measured by the expanded disability status scale (EDSS) (1.5 ± 1.4 vs. 2.1 ± 1.5, *p* = 0.001, baseline EDSS was 1.7 for both groups) and significantly lower activity on magnetic resonance imaging (MRI) (28% vs. 67%, *p* = 0.001) compared to the group on a regular diet [31]. There was no effect on annual relapse rate. Unfortunately, this study was subject to important limitations. Group allocation was performed by instructing all 72 patients to follow a GFD for the first six months of the study, whereafter non-compliant patients were asked to resume a regular gluten-containing diet. In addition, eight study participants were diagnosed with CD and they all remained in the GFD group, further supporting the idea that the method used for group allocation was suboptimal. Apart from above described methodological issues, major inconsistencies, including the use of the word “randomised” in the title of the study, reduce our capacity to trust this publication [31].

Eliminating gluten from the diet is also part of the “The Wahls Protocol”, a multimodal lifestyle intervention including, among others, adherence to a modified paleolithic diet. Clinical studies have illustrated that “The Wahls Protocol” can contribute to improving primarily self-reported outcomes, such as mood, fatigue and quality of life among patients with relapsing remitting MS [32] and progressive MS [33,34,35]. The risk of placebo and/or nocebo effects should not be neglected when evaluating the results of lifestyle interventions. Furthermore, due to the multimodal nature of the interventions, it is not possible to quantify the effects of eliminating gluten from the diet. The lack of disease-specific endpoints is a major limitation of these studies. Nevertheless, these publications highlight that lifestyle modifications can contribute to improving the quality of life of patients with MS. This is of utmost importance for patients with the progressive forms of MS, as highly effective treatments for these patients are still lacking [36].

### 3.2. Prevalence of Celiac Disease and Gluten-Related Serology in Multiple Sclerosis 

Several publications have reported the prevalence of gluten-related antibodies among patients with MS. Among six studies estimating the prevalence of seropositivity for anti-gliadin (AGA) immunoglobulins (Ig) in patients with MS [37,38,39,40,41,42], only one study found a significantly higher prevalence of IgG-AGA among patients with MS (7/98) compared to HCs (2/140) (*p* = 0.03) [42]. However, when investigating whether patients with MS have elevated mean values of IgA-AGA or IgG-AGA compared to HCs, the results are highly contradictive [38,42,43,44]. We can therefore not exclude that patients with MS may have slightly elevated AGA titers compared to HCs, however, this is still far from sufficiently different for diagnostic use.

Data from twelve studies [37,38,39,40,41,42,43,44,45,46,47,48] estimating the prevalence of seropositivity for IgA tissue transglutaminase (tTG) in patients with MS do not support an increased prevalence of CD among patients with MS, whereas a single study found higher mean values of IgA-tTG and IgG-tTG among 30 patients with MS compared to 25 HCs [49]. So far, only one publication supports an association between CD and MS by reporting the prevalence of CD to be 11% in a cohort of 72 patients with relapsing remitting MS and 32% (23/126) among their first-degree relatives [50]. According to the last-mentioned study, the diagnosis of MS was made at a younger age among celiac (35 +/− 7 years) compared to non-celiac (44 +/− 10 years) patients (*p* < 0.05) [50]. For an overview of studies measuring other gluten- and celiac-related antibodies among patients with MS, the reader is referred to Thomsen et al. (2019) [51].

The most powerful studies investigating a potential association between CD and MS are two Danish population-based studies [52,53] and a Swedish case–control study including 14,371 CD patients and 70,096 reference individuals [54], however, none of them found any association. The first Danish study investigated the comorbidity of 31 autoimmune diseases and calculated an odds ratio (OR) of 1.0 for CD and MS [52]. The second Danish study investigated the prevalence of autoimmune comorbidities among patients with CD and failed to find an increased prevalence of MS among patients with CD. According to the Swedish case–control study, the presence of CD did not increase the risk of subsequent MS diagnosis (hazard ratio (HR) = 0.9; 95% confidence interval (CI) = (0.3–2.3)) [54]. Lastly, two French studies estimated the prevalence of MS among patients with CD to be 0.11% (1/924) [55] and 0.14% (1/741) [56]. This is similar to the crude estimate of MS in the French population, which is 0.15% [57].

## 4. Psoriasis

Psoriasis is a chronic autoimmune skin disease characterized by the development of erythematous scaly lesions. Psoriasis vulgaris, also known as plaque psoriasis, is the most common type of psoriasis, but several other types of psoriasis also exist [58]. The Medical Board of the National Psoriasis Foundation conducted a systematic review in 2018 with the aim of developing nutritional recommendations for patients with psoriasis or psoriatic arthritis [59]. The board states “We weakly recommend a gluten-free diet only in patients who test positive for serologic markers of gluten sensitivity” [59]. The popularity of the GFD among patients with psoriasis is highlighted in a U.S. study from 2017, in which 38% of the responding patients (*n* = 1206) reported avoiding gluten and 53.4% (247/459) of them reported to have experienced an improvement or clearance of their disease as a result of the GFD [6].

### 4.1. Intake of Gluten and Risk of Psoriasis 

Using data from the Nurses´ Health Study II, a publication examined whether higher intakes of gluten were associated with a higher risk of future psoriasis, psoriatic arthritis and atopic dermatitis [60]. When comparing the highest and lowest gluten intake quintiles, the multivariate HRs were 1.15 (95%CI = (0.98–1.36)), 1.12 (95%CI = (0.78–1.62)) and 0.91 (95%CI = (0.66–1.25)) for psoriasis, psoriatic arthritis and atopic dermatitis, respectively. No dose–response relationship was observed, but the fact that the effect of a strictly GFD was not investigated is a minor limitation of this study. 

### 4.2. Gluten-Free Interventions in Psoriasis

The potential role of gluten in psoriasis has been addressed in several publications from Michaëlsson and his colleagues. In 2000, they published a study illustrating clinical improvement in 73% (22/30) of patients who adhered to a GFD for three months (reduction of psoriasis area and severity index (PASI) score from 5.5 ± 4.5 to 3.6 ± 3.0 (*p* = 0.001)) [61]. All patients were positive for IgA-AGA or IgG-AGA and no clinical improvement was observed among six seronegative patients who also adhered to a GFD. The study was originally designed as a cross-over trial and, after three months on a GFD, the participants had to reintroduce gluten to their diet for three months. However, the last part of the study was discontinued as 60% (18/30) of the AGA-positive patients, but none of the seronegative patients, required increased treatment due to a worsening of their skin lesions after the reintroduction of dietary gluten [61]. Immunohistochemical analyses of skin biopsies from 19 of the above seropositive patients were later published in a separate publication that revealed a reduction in Ki67 positive cells in the involved dermis after the GFD [62]. Moreover, a higher expression of tTG was found in involved, compared to uninvolved, dermis (5.06 ± 3.80% vs. 0.67 ± 0.54%, *n* = 13, *p* = 0.0002) and the GFD resulted in a drop in tTG expression in the dermis by 50% [62]. 

Similarly, in 2007 Michaëlsson et al. presented results from 16 cases of palmoplantar pustulosis, who adhered to a GFD [63]. AGA-seropositive patients who strictly adhered to the GFD (*n* = 9) experienced great improvements or even the clearance of their lesions. Improvements were only seen in two out of four patients with lower compliance to the GFD and none of the seronegative patients (*n* = 3) [63]. 

According to a more recent publication by Kolchak et al. in 2018, a one-year gluten-free intervention resulted in a 56% and 36% improvement in the PASI score in patients with very high (>30 U/ml, *n* = 5) and high (11.5–30.0 U/ml, *n* = 8) levels of IgA against gliadin peptides (not clear whether native or deamidated gliadin), respectively [64]. As no other antibodies were measured and biopsies were not performed, it is not known whether some of the included patients suffered from CD. The effects of a GFD in patients with concomitant psoriasis and CD have been explored in an Italian multicenter study [65]. At a three-month follow-up, patients (*n* = 9) experienced major improvements in their PASI scores (two by at least 50%, five by at least 75%, total clearance in one patient and one drop-out). A single patient had worsened by the six-month follow-up, whereas most patients maintained their clinical improvement (*n* = 5) and two patients further improved [65]. Overall, evidence suggests that psoriasis patients with gluten-related antibodies may benefit from a GFD, however, larger trials are still lacking.

### 4.3. Gluten-Related Serology in Psoriasis

According to a meta-analysis from 2014, both the prevalence of seropositivity for IgA-AGA, as well as mean values of IgA-AGA, are higher among patients with psoriasis compared to HCs [66]. To illustrate the heterogeneity among studies, an overview of identified case–control publications is provided in Table 1. Additional studies have described the prevalence of AGA seropositivity among patients with psoriasis [67,68] and palmoplantar pustulosis [63] without including a relevant control group. Moreover, most studies [69,70,71,72] have found significantly higher concentrations of IgA-AGA among patients with psoriasis compared to HCs, whereas one study [73] did not. Regarding IgG-AGA, only one [71] out of four studies [70,72,73] found higher concentrations in patients with psoriasis compared to HCs. Using a different approach, no difference was found in the proliferative response of peripheral blood mononuclear cells from patients with psoriasis (*n* = 37) and HCs (*n* = 37) after stimulation with wheat peptides, however, the five highest responses against peptide p62-75 were observed among patients with psoriasis [74].

To investigate whether gluten-related antibodies correlate with disease activity in psoriasis, a study screened 130 patients for IgG-AGA, IgA-AGA and IgA-tTG and identified 21 patients (16.2%) who were positive for at least one of the antibodies [80]. Psoralen and ultraviolet A (PUVA) phototherapy (57% vs. 30%, *p* = 0.03) and systemic therapy (48% vs. 22%, *p* = 0.04) was currently given or had previously been given to a higher percentage of seropositive compared to seronegative patients. There was no difference for ultraviolet B (UVB) phototherapy and the presence of arthritis or arthralgia [80]. A similar but smaller study (*n* = 41) found a significant relationship between seropositivity for IgA-AGA and disease duration (*p* < 0.001), however, being seropositive was not related to PASI scores [69]. In contrast, in a study with 120 patients with psoriasis (eight seropositive for IgA-AGA/five seropositive for IgG-AGA), severe disease at the reported time or past treatment for high disease severity was not associated with seropositivity for AGA [73]. 

According to a publication from 2020 that tested for antibodies against an array of 75 antigens, IgG4 antigliadin antibodies were the only antibodies to be elevated in the sera of 12 patients with severe psoriasis (PASI > 30) [81]. IgG4 antigliadin antibodies were not present in sera from 12 HCs. Later validation, using a cohort with 73 psoriasis patients and 75 HCs, resulted in an area under the curve of 0.98 (*p* < 0.001) in the receiver operating characteristic (ROC) analysis, suggesting that IgG4 antigliadin antibodies could potentially function as a diagnostic biomarker for psoriasis. For a subgroup of patients with the highest levels of anti-gliadin IgG4, there was a significant correlation between antibody levels and PASI scores (r = 0.65, *p* < 0.001) [81].

With regard to the prevalence of IgA-tTG antibodies in psoriasis, we have identified five cross-sectional cohort studies [63,68,80,82,83] and ten case–control studies [65,69,71,75,76,84,85,86,87,88] (previously mentioned in Table 2). In addition, a study of 67 patients with psoriasis and 85 HCs found significantly elevated mean values of IgA-tTG in patients with psoriasis (0.943 ± 1.131 vs. 0.852 ± 0.576, *p* < 0.05) [71]. Many case–control studies fail to reveal a significant difference between groups, however, it must be stressed that the majority of studies on the topic are underpowered, considering the fact that the global seroprevalence of CD has been estimated to be 1.4% [89]. 

### 4.4. Comorbidity between Celiac Disease and Psoriasis

Results from 18 publications included in a systematic review and meta-analysis from 2019 are summarized below [90]. Out of two studies investigating the incidence of CD among patients with psoriasis, only one found a statistically significant increased risk (HR = 1.9, 95%CI = (1.6–2.2) [91] and HR = 1.20, 95%CI = (0.91–1.59)) [92]. Similarly, two studies estimated the incidence of psoriasis among patients with CD, but in this case, both studies found significant results (HR = 1.72, 95%CI = (1.54–1.92) [93] and HR = 1.9, 95%CI = (1.5–2.3) [91]). With regard to the prevalence of CD in patients with psoriasis or psoriatic arthritis, significantly increased ORs were found in five [65,92,94,95,96] out of nine [86,87,97,98] studies (meta-analysis: OR = 2.16, 95%CI = (1.74–2.69) [90]). Likewise, the prevalence of psoriasis among patients with CD was found to be increased in four [53,93,99,100] out of eight [101,102,103,104] publications (meta-analysis: OR = 1.8, 95%CI = (1.36–2.38) [90]). The two-way meta-analysis concluded that clinicians should be aware of the significant association between CD and psoriasis [90].

## 5. Type 1 Diabetes 

T1D is a chronic, autoimmune disease characterized by the destruction of the insulin-producing beta cells in the pancreas. T1D is often diagnosed in childhood and results in a lifelong need for exogenous insulin. Several animal studies support the potential involvement of gluten in the pathogenesis of T1D [105] and have previously been summarized by Antvorskov et al. [105] and Haupt-Jorgensen et al. [106].

### 5.1. Exposure to Gluten during Early Life and Risk of Type 1 Diabetes

Recent mother and child cohort studies suggest that exposure to gluten during early life may affect the risk of developing T1D [107,108], whereas earlier studies among predisposed individuals did not reveal such an association [109,110]. According to a Danish study from 2018 [107], offspring from mothers with the highest intake of gluten had a twofold higher risk of developing T1D compared to offspring from mothers with the lowest intake of gluten during pregnancy (adjusted HR = 2.00, 95%CI = (1.02–4.00)). A dose–response relationship was demonstrated, however, only the difference between the groups with the highest and lowest intakes of gluten reached statistical significance. These results were not replicated in a similar Norwegian study from 2020 [108] but, this time, the intake of gluten by the offspring themselves was associated with a higher risk of T1D (adjusted HR = 1.46, 95%CI = (1.06–2.01), *p* = 0.02). 

Similarly, publications have explored whether and how infant feeding patterns could affect the risk of T1D [105]. Data from the Diabetes Autoimmunity Study in the Young (DAISY) support that introduction of cereals between the age of 4-6 months leads to the lowest risk of islet autoimmunity (<4 months: HR = 4.32, 95%CI = (2.0–9.35), >6 months: HR = 5.36, 95%CI = (2.08–13.8)) [111]. Likewise, the late (≥7 months) introduction of gluten-containing porridge has been found to be a risk factor for the development of β-cell autoantibodies [112], whereas Ludvigsson [113] did not find an association between the time of the introduction of gluten and levels of islet autoantibodies. Results from the BABYDIAB study support the idea that the introduction of gluten-containing foods at or before the age of three months increases the risk of islet autoimmunity (HR = 5.2, 95%CI = (1.7–15.5), *p* = 0.003), however, the late (>6 months) introduction of gluten-containing foods was not associated with increased risk of islet autoimmunity [114]. Lastly, the BABYDIET study—a pilot study in which 150 infants at high risk of T1D where randomized to either control (6 months) or late (12 months) introduction of gluten—did not find any difference in islet autoimmunity at three years in the per protocol analysis (compliance = 70%) [115].

### 5.2. Gluten-Free Interventions in Type 1 Diabetes

Two studies have investigated whether a GFD could have a protective effect among children with a high risk of developing T1D. In the first study, 17 first-degree relatives of T1D patients with at least two β-cell autoantibodies were included in a cross-over trial consisting of six months on a GFD followed by six months on a gluten-containing diet [116]. Glucose tolerance tests revealed an improved acute insulin response after the GFD (*p* = 0.004) and a non-significant deterioration after the reintroduction of dietary gluten (*p* = 0.07). The results were similar for insulin sensitivity measured by the homeostasis model of insulin resistance (HOMA-IR), however, this time, a non-significant increase after the GFD was followed by a significant decrease (*p* < 0.005) after the gluten-containing diet. An effect on the levels of autoantibodies was neither observed in this study [116] nor in a similar study with a longer gluten-free intervention of 12 months (*n* = 7) [117]. A five-year follow-up to the latter study suggests that the 12 months on a GFD did not affect the risk of progressing to T1D [117]. 

In 2012, a case report suggested that a GFD introduced 2–3 weeks after the diagnosis of T1D may have prolonged remission in a five-year-old boy without CD [118]. Both his HbA1c and his fasting blood glucose stabilized without insulin therapy and, twenty months after diagnosis, he still remained without the need for daily insulin therapy [118]. This case led to the performance of a Danish pilot study evaluating the effects of a one-year gluten-free intervention among 15 children with newly diagnosed T1D [119]. Compared to two previous reference cohorts, the children on the GFD had a 21% lower HbA1c and a higher prevalence of partial remission (insulin dose-adjusted A1c (IDAA1c) ≤ 9), but no difference was seen in stimulated C peptide [119]. 

Prolonged partial remission in response to a GFD was also illustrated in a study that was methodologically stronger due to the inclusion of a control group which remained on a standard gluten-containing diet (*n* = 19) during the study time [120]. The trial was not randomized and 20 out of 26 children completed the one-year gluten-free intervention with satisfactory compliance. The GFD was introduced within a median of 38 days from the onset of T1D. At follow-up, the children adhering to the GFD had a lower IDAA1c (by 1.37; *p* = 0.01), a lower mean HbA1c (by 0.7% (7.8 mmol/mol); *p* = 0.02) and there was a tendency towards a lower insulin dose (by 0.15 U/kg/day; *p* = 0.07) compared to the control group [120]. Last, but not least, several studies have investigated whether a GFD affects metabolic control in individuals with both CD and T1D [121,122,123,124,125,126,127,128,129,130,131,132,133,134,135,136,137,138,139,140,141,142] (presented in Table A1 in Appendix A). 

### 5.3. Prevalence of Celiac Disease and Gluten-Related Serology in Type 1 Diabetes

A systematic review and meta-analysis published in 2014 calculated the prevalence of biopsy-confirmed CD among patients with T1D to be 6.0% (95%CI = (5.0–6.9%)) [143]. Similarly, a systematic review and meta-analysis from 2019 reports the weighted prevalence of CD and any gluten-related antibodies among patients with T1D to be 4.7% (95%CI = (4.0–5.5)) and 10.2% (95%CI = (8.4–12.7)), respectively [144]. Among gluten-related antibodies, the highest weighted prevalence was estimated for IgG-AGA as 12.7% (95%CI = (6.1–21.0)) [144]. Equally relevant, a Swedish population cohort study has estimated the HR of subsequent T1D before the age of 20 years to be 2.4 (95%CI = (1.9–3.0), *p* < 0.001) among patients with CD [145].

Although the association between CD and T1D is well supported, the heterogeneity among studies is large. A better understanding of the factors that contribute to this variation may therefore be relevant. With regard to measurements of gluten-related antibodies, technical differences in the analytical assays being used may hinder direct comparisons among publications highlighting the importance of including a healthy control group in all studies. This is especially relevant for AGA due to their lower specificity for CD and the fact that biological factors may contribute to their variation within healthy populations. 

The meta-analysis by Elfström et al. [143] revealed that CD was less frequent in adults (2.7%, 95%CI = (2.1–3.3%)) compared to children (6.2%, 95%CI = (6.1–6.3%)) with T1D (*p* < 0.001). Tiberti et al. [146] on the contrary, found a significantly higher prevalence of gluten-related antibodies among patients with a high (> 15 years) compared to a low (5–15 years) duration of T1D. Similarly, Nederstigt et al. [144] reported that the prevalence of IgA-AGA increased with the duration of T1D, whereas endomysium antibodies decreased with age. Interestingly, IgA-tTG-seropositive patients with T1D have been found to have lower titers of IgG-tTG and deamidated gliadin peptide antibodies compared to CD patients without T1D [147]. In addition, longitudinal studies suggest that AGA titers can fluctuate over time [148] but also that diabetes-related antibodies may respond to a GFD in cases with CD [149]. Lastly, data from Salardi et al. [150] support that the prevalence of CD significantly increased among Italian patients with T1D after 1994, however, this might also reflect an increase in the prevalence of CD in the general population.

## 6. Autoimmune Thyroid Diseases

ATDs affect 2–5% of the population with a female predominance. The most common ATDs are Hashimoto’s thyroiditis (HT) and Graves’ disease, which lead to hypothyroidism and hyperthyroidism, respectively [114].

### 6.1. Gluten-Free Interventions in Autoimmune Thyroid Diseases

Few studies have investigated whether a GFD can contribute to ameliorating thyroid-related pathology among patients with concomitant CD, but we have not been able to identify publications exploring the effects of a GFD in ATDs in the absence of CD or celiac-related antibodies. 

A controlled trial has investigated the effects of six months on a GFD (*n* = 16) compared to no dietary intervention (*n* = 18) among drug-naive women with HT [151]. The GFD resulted in a drop in the levels of thyroid peroxidase (TPO) and thyroglobulin antibodies, an increase in 25-hydroxyvitamin D and an improvement in the structure parameter inference approach (SPINA)-GT index, which correlated with the changes in antibody titers. No effect was seen on levels of thyrotropin and free triiodothyronine. The study population included patients that were seropositive for IgA-tTG, however, no intestinal biopsies were performed and patients with symptomatic CD were excluded [151].

An Italian multicenter study evaluating the thyroid function of 128 patients with newly diagnosed CD before and one year after the introduction of a GFD reports that, in some patients, a GFD can reverse thyroid abnormalities [152]. Valentino et al. [153] also noted an improvement in symptoms related to hypothyroidism and thyroxine dosage among three ATD patients with concomitant CD that followed a GFD for six months. However, levels of thyroglobulin and TPO antibodies only changed for one patient, who had an additional follow-up at 18 months [153]. 

On the contrary, Mainardi et al. [154] report that a GFD did not seem to influence thyroid function and antibodies among two cases of concomitant CD and ATD. Likewise, a more recent study found no effect of one year on a GFD on levels of TPO antibodies that were present among 10 (37%) patients with newly diagnosed CD [155]. On the contrary, thyroid volume significantly decreased compared to a group of patients without CD, indicating that thyroiditis was continually progressing even after the establishment of a GFD [155]. It is possible that a longer study time is necessary to reveal an effect of a GFD, as TPO antibodies were only present among 76.9% (10/13), 46.1% (6/13) and 15.3% (2/13) of CD patients with ATD at, respectively, 6-, 12- and 24-month follow-ups on a GFD [149].

Interestingly, a study found that patients with concomitant CD and HT (*n* = 14) needed an almost 50% higher dose of levothyroxine to reach target thyroid-stimulating hormone (TSH) values when compared to patients with HT alone (*n* = 68) [156]. The authors suggest that this could potentially be explained by reduced absorption of levothyroxine in cases of untreated CD, as an increased need for levothyroxine was prevented by the introduction of a GFD (*n* = 21). However, reduced absorption capacity cannot explain why patients with concomitant HT and CD had significantly higher TSH (5.7 vs. 7.26, *p* = 0.0099) and significantly lower free T4 (1.12 vs. 0.01, *p* < 0.0001) compared to patients with isolated HT before the initiation of levothyroxine treatment [156]. In accordance with the above, Zubarik et al. [157] reported that patients requiring high doses of levothyroxine to maintain an euthyroid state were more likely to have CD, but this was not confirmed by Sharma et al. [158]. 

### 6.2. Gluten-Related Serology in Autoimmune Thyroid Diseases

Identified studies measuring levels of IgA-AGA and IgG-AGA in ATDs are summarized in Table 3. Furthermore, a study measuring the presence of IgG antibodies against 125 foods found no difference in IgG positivity for wheat or gliadin between 74 patients with HT and 245 HCs [180], but IgG positivity for barley was significantly higher among patients with HT compared to HCs (93.2% vs. 71.0%, *p* = 8.4 × 10^−5^) [180]. Moreover, a study supporting the previously discussed association between CD and an increased need for levothyroxine found that patients treated with high dosage of levothyroxine (125–200 μg/day) had significantly higher levels of IgA-AGA compared to patients receiving low levels of levothyroxine (50–100 μg/day) (medians: 19.69 vs. 13.00, *p* = 0.033) [162]. 

An interesting study found that the prevalence of chronic thyroiditis or seropositivity for TPO antibodies was higher among 16 patients with T1D that were seropositive for AGA compared to 37 AGA-seronegative T1D patients (38% vs. 2.7%, *p* = 0.005 for chronic thyroiditis and 69% vs. 27%, *p* = 0.01 for TPO seropositivity) [181]. This is further supported by a study reporting that the prevalence of tTG (*p* = 0.023) and glutamic acid decarboxylase (GAD) (*p* < 0.00001) antibodies increased with increasing titers of TPO antibodies [164]. A correlation between TPO and IgA-tTG antibodies has also been illustrated in a study suggesting that IgA-tTG may contribute to thyroid dysfunction by binding to thyroid tissue [182]. Similarly, IgG-tTG and IgA-AGA have been found to be predictors of TPO and thyroglobulin antibodies, respectively (IgG-tTG/ TPO: β = 0.12, 95%CI = (0.03–0.21), *p* = 0.008, IgA-AGA/thyroglobulin: β = −0.10, 95%CI = (−0.19–−0.002), *p* = 0.045) [168]. The association between ATDs, T1D and CD has been confirmed by additional publications [178,183], including a population-based cohort study concluding that CD is a risk factor for later development of ATDs among patients with T1D [184]. 

### 6.3. Comorbidity between Celiac Disease and Autoimmune Thyroid Diseases

A systematic review and meta-analysis of 27 studies calculated the median prevalence of CD in ATDs to be 3.2%, however, a pooled analysis resulted in a prevalence of 1.6% (CI = (1.3–1.9%)) for biopsy-verified CD [185]. The abovementioned low prevalence can possibly be explained by the fact that intestinal biopsies are not performed in all seropositive patients with potential CD. Furthermore, the prevalence of CD was higher among children with ATDs (6.2%, CI = (4.0–8.4%)) compared to adults (2.7%, CI = (2.1–3.4)) [185], whereas another study suggests that the prevalence of CD is higher among patients with ATDs above the age of 65 [167].

A meta-analysis of a systematic review from 2016 revealed a significantly higher prevalence of thyroid disease among patients with CD compared to controls (OR = 3.08, 95%CI = (2.76–3.56)) [186]. Similar results were also found for euthyroid ATD (OR = 4.35, 95%CI = (2.88–6.56)) and hypothyroidism (OR = 3.38, 95%CI = (2.73–4.19)), however, the prevalence of hyperthyroidism among patients with CD did not differ from that in controls (OR = 1.28, 95%CI = (0.37–4.46)) [186]. On the contrary, data from 3209 patients with Grave’s disease and 1069 HCs support the idea that the prevalence of CD is higher among patients with Grave’s disease (1.1%) compared to HCs (0.3%) (OR = 3.81, 95%CI = (1.17–12.41)) [187]. Additionally, a meta-analysis reports that the prevalence of biopsy-proven CD is higher among patients with hyperthyroidism (2.6%, CI = (0.7–4.4%)) compared to patients with hypothyroidism (1.4%, CI = (1.0–1.9%)) [185]. We hypothesize that the late age of disease onset for hyperthyroidism could be a potential explanation for the above contradictive results. An association between thyroid disease and CD has also been confirmed by more recent studies [188,189]. One calculated that the prevalence of thyroid disease was fourfold higher among 288 patients with untreated CD compared to 250 controls without CD (13.6% vs. 3.2%, *p* < 0.05) [188] and the other calculated the hazard ratio of subsequent hypothyroidism among patients with CD to be 4.64 (95%CI = (2.88–7.46)) [189].

It has been debated whether the late diagnosis of CD and, as a result, the late introduction of a GFD can increase the risk of developing other autoimmune diseases [55,102,103,190]. When a meta-analysis compared treated and untreated patients with CD, no difference was found in the frequency of thyroid disease (OR = 1.08, 95%CI = (0.61–1.92)) [186]. In addition, a study highlights that first-degree relatives of patients with CD also have an increased risk of ATDs [191]. Interestingly, a study reports that the prevalence of ATDs among Irish women with CD has decreased significantly over recent decades [192], whereas another study suggests that the prevalence of autoimmune thyroiditis may be higher among seronegative (26.9%) compared to seropositive (9.7%) patients with CD (*p* = 0.002) [193]. Last, but not least, the prevalence of ATDs has also been reported to be high among patients with non-celiac gluten/wheat sensitivity [194,195] and dermatitis herpetiformis [196].

## 7. Conclusions

The current level of evidence is yet not sufficient to recommend a GFD to patients with MS, psoriasis, T1D or ATDs. Larger epidemiological studies and meta-analyses of systematic reviews support that psoriasis, T1D and ATDs are all associated with CD, but this does not seem to be the case for MS. The only clinical trial to have studied the effects of a GFD among patients with MS found positive results on important MS-specific outcomes, however, the publication was subject to major limitations. Further studies are warranted to replicate the results found by Rodrigo et al. [31] and clarify whether any beneficial effects could be restricted to specific subgroups of patients. With regard to psoriasis, the currently available data suggest that patients with gluten-related antibodies or CD may benefit from a GFD, however, larger trials are still missing. The majority of studies failed to reveal an effect of a GFD on diabetes-related autoantibodies, however, it seems likely that a GFD may contribute to normalizing metabolic control in patients with T1D. In addition, some publications report that untreated CD can affect metabolic control and diabetic complications in patients with T1D. On the contrary, studies support the idea that thyroid-related antibodies may respond to a GFD in patients with concomitant CD and ATD, however, no studies have addressed the effects of a GFD among non-celiac patients with ATDs to date. Lastly, in patients with concomitant CD and ATD, a GFD may improve the absorption of levothyroxine.

## Figures and Tables

**Table 1 nutrients-12-02316-t001:** Case–control studies estimating the prevalence of IgA-AGA and IgG-AGA in patients with psoriasis and HCs. Results are presented as “number of seropositive individuals”/“number of individuals tested”. HCs: healthy controls, Ig: immunoglobulin, AGA: antigliadin antibody, NA: not available, NS: not significant.

Case–Control Studies	IgA-AGA in Psoriasis	IgA-AGA in HCs	*p*-Value	IgG-AGA in Psoriasis	IgG-AGA in HCs	*p*-Value
Akbulut 2013 [75]	6/37	1/50	*p* = 0.039	3/37	0/50	*p* = 0.073
Cardinali 2002 [76]	0/39	0/39	NS	2/39	0/39	NA
Kalayciyan 2006 [77]	21/127	3/31	NS	-	-	-
Kia 2007 [78]	6/200	5/100	NS	32/200	16/100	NS
^1^ Kolchak 2018 [64]	13/97	2/91	NA	-	-	-
Lesiak 2016 [79]	0/20	0/29	NS	-	-	-
Nagui 2010 [69]	14/41	1/41	NA	-	-	-
Singh 2010 [71]	8/56	0/60	*p* < 0.05	12/56	0/60	*p* < 0.05
Sultan 2010 [73]	8/120	9/120	NS	5/120	6/120	NS
In total	76/737	21/561	-	54/452	22/369	-

^1^ Possibly measures of antibodies against deamidated gliadin peptide.

**Table 2 nutrients-12-02316-t002:** Case–control studies estimating the prevalence of IgA-tTG in patients with psoriasis and HCs. Results are presented as “number of seropositive individuals”/“number of individuals tested”. HCs: healthy controls, IgA-tTG: class A immunoglobulins against tissue transglutaminase, NA: not available, NS: not significant.

Case–Control Studies	IgA-tTG in Psoriasis	IgA-tTG in HCs	*p*-Value
Akbulut 2013 [75]	1/37	0/50	NS
Bastiani 2015 [65]	9/218	1/264	*p* < 0.05
Cardinali 2002 [76]	1/39	0/39	NS
Hull 2008 [84]	0/37	1/53	NS
Juzlova 2016 [85]	2/189	0/378	*p* = 0.045
Montesu 2011 [86]	2/100	0/100	NS
Nagui 2010 [69]	14/41	9/41	NA
Ojetti 2003 [87]	2/92	0/90	NA
Riente 2004 [88]	1/75	3/78	NS
Singh 2010 [71]	6/56	0/60	*p* = 0.01
In total	38/884	14/1153	-

**Table 3 nutrients-12-02316-t003:** Case–control and cross-sectional cohort studies estimating the prevalence of IgA-AGA and IgG-AGA in patients with ATD and HCs. Results are presented as “number of seropositive individuals”/“number of individuals tested” (%). AGA: antigliadin antibody, ATD: autoimmune thyroid disease, HCs: healthy controls, Ig: immunoglobulin, NA: not available.

Study	IgA-AGA in ATD	IgA-AGA in HCs	*p*-Value	IgG-AGA in ATD	IgG-AGA in HCs	*p*-Value	IgA-tTG in ATD	IgA-tTG in HCs	*p*-Value
Ch’ng 2005 [159]	15/111 (13.5%)	-	-	-	-	-	2/111 (1.8%)	1/115 (0.9%)	NA
Guliter 2007 [160]	-	-	-	-	-	-	8/136 (5.8%)	1/119 (0.8%)	*p* = 0.04
Hadithi 2007 [161]	9/104 (8.7%)	-	-	7/104 (6.7%)	-	-	8/104 (7.7%)	-	-
Jiskra 2003 [162]	27/169 (16.0%)	101/1312 (7.7%)	*p* = 0.002	87/169 (51.5%)	92/1312 (7.0%)	*p* < 0.001	25/169 (14.8%)	-	-
Mainardi 2002 [154]	-	-	-	-	-	-	2/100 (2%)	-	-
Mankai 2006 [163]	-	-	-	-	-	-	6/161 (3.7%)	-	-
Marwaha 2013 [164]	-	-	-	-	-	-	40/577 (6.9%)	20/577 (3.5%)	*p* = 0.015
Mehrdad 2012 [165]	3/454 (0.7%)	-	-	-	-	-	8/454 (1.8%)	-	-
Meloni 2000 [166]	13/297 (4.4%)	-	-	18/297 (6.1%)	-	-	-	-	-
Ravaglia 2003 [167]	-	-	-	46/737 (6.2%)	7/600 (1.2%)	NA	11/737 (1.5%)	2/600 (0.3%)	*p* = 0.046
Riseh 2017 [168]	6/40 (15.0%)	5/42 (11.9%)	NA	2/40 (5.0%)	4/42 (9.5%)	NA	9/40 (22.5%)	7/42 (16.6%)	NS
Sahin 2018 [169]	-	-	-	-	-	-	3/66 (4.6%)	-	-
Sari 2009 [170]	-	-	-	-	-	-	8/101 (7.9%)	0/103 (0.0%)	NA
Sattar 2011 [171]	-	-	-	-	-	-	14/302 (4.6%)	-	-
Sharma 2016 [158]	-	-	-	-	-	-	24/280 (8.6%)	-	-
Spadaccino 2008 [172]	-	-	-	-	-	-	10/271 (3.7%)	-	-
Tuhan 2016 [173]	-	-	-	-	-	-	1/80 (1.3%)	-	-
Twito 2018 [174]	-	-	-	-	-	-	5/114 (4.4 %)	-	-
Valentino 2002 [175]	0/14 (0.0%)	-	-	0/14 (0.0%)	-	-	0/14 (0.0%)	-	-
Ventura 2014 [176]	-	-	-	-	-	-	2/53 (3.8%)	-	-
Volta 2001 [177]	-	-	-	6/220 (2.7%)	3/250 (1.2%)	NA	7/20 (3.2%)	1/250 (0.4%)	*p* = 0.022
Zhao 2016 [178]	-	-	-	-	-	-	26/119 (21.9%)	1/102 (1.0%)	*p* < 0.0001
Zubarik 2015 [179]	-	-	-	-	-	-	10/499 (2.0%)	-	-

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
