# Peer review of "Current Evidence on the Efficacy of Gluten-Free Diets in Multiple Sclerosis, Psoriasis, Type 1 Diabetes and Autoimmune Thyroid Diseases"

_nutrients, 2020, doi:10.3390/nu12082316_

Round 1
Reviewer 1 Report
This is a good piece of work gathering all the evidence about Gluten-Free Diet in diseases other than celiac disease. It is a very interesting topic that needs further research.
Some minor changes are required:
- I think that introduction should address all the diseases the authors are intended to review. Diabetes and ATD are lacking.
- L 85. Please specify what MRI means.
- Correct (22/30)
- It can be a little misleading to mention treatment need in a cohort of patients who previously had reported a clinical improvement. I suggest to delete the sentence “Crossover to a gluten-containing diet was attempted but not completed due to the need for increased treatment by 60% (18/30) of the AGA positive patients [62]".
- I suggest to introduce the new study [63], it seems that you are still analyzing the previous one [62].
- Table1 and L165. What does footnote mean? Should we assume that rest of the studies measured native gliadin? Why did you suspect that Kolchak et al are using deamidated gliadin IgA? It should be clearly stated in the M&M section of this paper, and they did not specify that it is deamidated, did they? Please clarify.
- Correct: Zubarik et al reportED
- L 385. Correct: has also been illustrated in A study…
- Could you please specify why you did not cover 189 and 190 studies in 6.3 section?
Author Response
Thank you very much for your valuable comments. Please see our answers below.
I think that introduction should address all the diseases the authors are intended to review. Diabetes and ATD are lacking.
We have added following sentences to the introduction:
Type 1 diabetes (T1D) and autoimmune thyroid diseases (ATD) affect the endocrine system. The contribution of dietary factors to the pathogenesis of autoimmune endocrine disorders is currently an active research area.
L 85. Please specify what MRI means.
Corrected to: significantly lower activity on magnetic resonance imaging (MRI) (28% vs 67%, p=0.001) compared to the group on a regular diet
Correct (22/30)
It is not clear what we should correct. We have removed spacing between “/” and “30”.
It can be a little misleading to mention treatment need in a cohort of patients who previously had reported a clinical improvement. I suggest to delete the sentence “Crossover to a gluten-containing diet was attempted but not completed due to the need for increased treatment by 60% (18/30) of the AGA positive patients [62]".
We still believe that above point is essential, but we agree that the sentence may be misleading. We have tried to clarify our point by inserting below text instead:
The study was originally designed as a cross-over trial and after three months on a GFD, participants had to reintroduce gluten to their diet for three months. However, the last part of the study was discontinued as 60% (18/30) of the AGA positive patients, but none of the seronegative patients, required increased treatment due to worsening of their skin lesions after reintroduction of dietary gluten.
I suggest to introduce the new study [63], it seems that you are still analyzing the previous one [62].
This sentence does indeed present a new publication (63), however, publication 63 is not independent from publication 62. Publication 63 presents additional findings generated from the study population described in publication 62. We have added following underlined words to our manuscript:
Immunohistochemical analyses of skin biopsies from 19 of above seropositive patients were later published in a separate publication that revealed a reduction in Ki67 positive cells in the involved dermis after the GFD [63]
Table1 and L165. What does footnote mean? Should we assume that rest of the studies measured native gliadin? Why did you suspect that Kolchak et al are using deamidated gliadin IgA? It should be clearly stated in the M&M section of this paper, and they did not specify that it is deamidated, did they? Please clarify.
AGA is commonly used as the short form for anti-gliadin antibodies. AGA - contrary to DGP (deamidated gliadin peptide) antibodies - have been raised against gliadin peptides that have not been subjected to modification by the enzyme tissue transglutaminase. AGA antibodies are more prevalent and have lower specificity for celiac disease compared to DGP antibodies.
It is unclear to us whether Kolchak et al. have measured IgA-DGP or IgA-AGA as the short form AGA is used in the publication, however, following sentence is used to describe their method:
“Antigliadin levels were assessed by the EliA IgA GliadinDP 250 Method of Phadia…”
According to the provider´s webpage the EliA IgA GliadinDP 250 Method of Phadia uses synthetic deamidated gliadin peptides as antigens.
https://www.thermofisher.com/phadia/wo/en/product-catalog.html?articleNumber=14-5538-01®ion=GR
Correct: Zubarik et al reported
Corrected
L 385. Correct: has also been illustrated in A study…
Corrected
Could you please specify why you did not cover 189 and 190 studies in 6.3 section?
We did not cover studies 189 and 190 in detail as they support the conclusions of the previously presented meta-analysis without adding new information that is relevant for this review. However, we still chose to provide the references for studies 189 and 190 as they were published after the cited meta-analysis and therefore not included in its calculation. We have now added the following sentence to our manuscript.
An association between thyroid disease and CD has also been confirmed by more recent studies [189,190]. The one calculated that the prevalence of thyroid disease was 4-fold higher among 288 patients with untreated CD compared to 250 controls without CD (13.6% vs 3.2%, p<0.05) [189] and the other calculated the hazard ratio of subsequent hypothyroidism among patients with CD to be 4.64 (95%CI=(2.88-7.46)) [190].
Reviewer 2 Report
I liked very much the review, congratulations!.
Please review English grammar at line 21.
I would like to have a major coherence between the abstract and conclusions. I agree with your conclusions, but the abstract appears a little more positive than your conclusions.
Author Response
We are happy to read that you liked our paper.
We have tried to address your wish by adjusting our abstract. Please see the revised version below:
In this review, we summarize the clinical data addressing a potential role for gluten in multiple sclerosis (MS), psoriasis, type 1 diabetes (T1D) and autoimmune thyroid diseases (ATD). Furthermore, data on the prevalence of celiac disease (CD) and gluten-related antibodies in the above patient groups are presented. Adequately powered and properly controlled intervention trials investigating the effects of a gluten-free diet (GFD) in non-celiac patients with MS, psoriasis, T1D or ATD are lacking. Only one clinical trial has studied the effects of a GFD among patients with MS finding promising results. A few publications have found beneficial effects of a GFD in a subgroup of patients with psoriasis that were seropositive for anti-gliadin or deamidated gliadin antibodies, but no effects were seen among seronegative patients. Studies on the role of gluten in T1D are contradicting, however, it seems likely that a GFD may contribute to normalizing metabolic control without affecting levels of islet autoantibodies. Lastly, the effects of a GFD in non-celiac patients with ATD have not been studied yet, but some publications report that thyroid-related antibodies respond to a GFD in patients with concomitant CD and ATD. Overall, there is currently not enough evidence to recommend a GFD to non-celiac patients with MS, psoriasis, ATD or T1D.
Reviewer 3 Report
The paper summarizes studies exploring the effect of a gluten-free diet on multiple sclerosis, psoriasis, type 1 diabetes and autoimmune thyroid disease. It is well-written in general, understandable and discusses the relevant papers. There are only some minor points to correct.
L45: Please insert wheat here: baking quality of wheat flours. Gluten is also found in rye and barley, but is not of relevance for baking quality in those two cereals.
L52: What about the glutelin fraction of these cereals?
L95 and in general: What about nocebo and placebo effects? These were found the be quite significant in many studies.
Author Response
Thank you very much for your comments. We agree with your suggestions and have made following corrections to our manuscript:
L45: Please insert wheat here: baking quality of wheat flours. Gluten is also found in rye and barley, but is not of relevance for baking quality in those two cereals.
The word wheat has been added.
L52: What about the glutelin fraction of these cereals?
Following sentence has been added: Likewise, the glutelin fractions of rye and barley are commonly described as secalinin and hordenin, however, similar terminology does not apply for oat glutelins [12].
L95 and in general: What about nocebo and placebo effects? These were found the be quite significant in many studies.
Following sentence has been added to the section describing the studies on “The Wahls Protocol”: The risk of placebo and/ or nocebo effects should not be neglected when evaluating the results of lifestyle interventions.